

# Integrated bioinformatics screening and experimental validation: construction of a LUAD prediction model based on Treg-related genes

Tian Zhao[1], Yan Yao[1,2], Yan Sun[1], Qingliang Lv[3], Changgang Sun[1,2], Yining Cheng[4], Chundi Gao[1] and Jing Zhuang[2]

[1] College of Traditional Chinese Medicine, Shandong Second Medical University, Weifang, China
[2] Department of Oncology, Weifang Traditional Chinese Hospital, Weifang, China
[3] Department of Interventional Radiology, Weifang People's Hospital, Weifang, China
[4] College of First Clinical Medicine, Shandong University of Traditional Chinese Medicine, Jinan, China

Corresponding authors
Chundi Gao,
gaochundi2017@163.com
Jing Zhuang, 13963676719@163.com

## ABSTRACT

**Background**. The prognosis of lung adenocarcinoma (LUAD) is poor, and clinical treatment mainly comprises a combination of traditional therapy and immunotherapy. However, the role and mechanism of tumor-infiltrating regulatory T cells (Tregs) in immunotherapy remain controversial. Therefore, we aimed to determine the role of Tregs in LUAD and to construct a relevant prognostic model for future clinical treatment.

**Methods**. A LUAD dataset was downloaded from the Gene Expression Omnibus (GEO) database, screened, integrated, and divided into test and validation datasets. CIBERSORT and weighted correlation network analysis (WGCNA) algorithms were combined to screen for Treg cell-related modules. Minimum absolute contraction and selection operator (LASSO) and univariate and multivariate Cox regression analyses were used to screen genes in the key modules and construct Treg-related prognostic models. Then, the expression differences of genes in the prognostic model were analyzed, and the results were verified by Western blotting.

**Result**. Among all cluster modules, the correlation between the brown module and Tregs ($R^2 = 0.43$, $P = 1e - 30$) was the highest. After LASSO and univariate and multivariate Cox regression screening, six genes (*ADARB2*, *B3GALT*, *FER*, *LTB4R2*, *N6AMT1*, and *SCN9A*) were selected to construct the prognosis model, and the prognosis of low-risk patients was found to be better than that of high-risk patients. Finally, the *SCN9A* gene with the highest correlation with the model was selected and verified using Western blot analysis. The results showed that the expression of Treg surface markers in LUAD cells was increased, and the expression of *SCN9A* was decreased compared with that in normal lung epithelial cells.

**Conclusion**. We identified the role of Treg-related genes in LUAD, constructed and verified a related prognostic model, and explored a potential therapeutic target, SCN9A, to provide a new perspective for the clinical treatment of LUAD.

## BACKGROUND

Lung adenocarcinoma (LUAD) is a non-small cell lung cancer (NSCLC) that develops in the small airway gland cells on the outer edge of the lung. Lung cancer is the most common type of cancer worldwide. It accounts for approximately 40% of all lung cancer cases (*Sung et al., 2021*; *Testa, Castelli & Pelosi, 2018*; *Cao et al., 2020*; *Ferlay et al., 2018*). LUAD is mainly treated using multidisciplinary comprehensive treatment, and traditional surgery and chemoradiotherapy remain the preferred mode. However, owing to the high heterogeneity and drug resistance of LUAD, only few patients respond well to these conventional treatments, and the overall prognosis of patients with lung cancer is poor, with a 5–20% survival rate (*Hirsch et al., 2017*). Immunotherapy has shown great progress in LUAD (*Fehrenbacher et al., 2016*; *Gower & Garon, 2022*; *Blumenschein Jr et al., 2015*; *Jänne et al., 2013*), and the prognosis of patients has been greatly improved by the combination of immunotherapy and traditional treatment methods (*Gradishar et al., 2021*; *Clarke, 2015*; *Blattman & Greenberg, 2004*; *Huang et al., 2024*). Therefore, understanding the relationship between the immune system and tumor prognosis, and exploring potential therapeutic targets, are of great importance for future LUAD treatments (*Prabhakaran et al., 2017*).

The tumor microenvironment (TME) is closely related to cancer prognosis. The TME refers to the self-sufficiency of tumor cells through the recruitment of other invasive immune cell populations, thus constituting a complete biological unit (*Kemi et al., 2018*; *Peltanova, Raudenska & Masarik, 2019*). It is mainly composed of different immune cells such as B cells, CD4$^+$ helper T cells, and regulatory T cells (Tregs), and thus determines the performance characteristics of tumors and the prognosis of patients (*Bremnes et al., 2016*). In some cancers, immune cells are closely associated with tumor growth, invasion, and metastasis (*Pagès et al., 2010*; *Domingues et al., 2016*). Among them, Tregs have become a key player in the TME and are closely related to the prognosis of the tumor (*Zhang et al., 2023*).

Tregs, a unique cluster of CD4$^+$ CD25$^+$T cells, are a type of T cells that mainly express CD4 molecules on their cell surface. Tregs can be roughly divided into the following types according to their production sites: natural Treg cells (NTregs), produced in the thymus and are the most common; adaptive Treg cells (PTreg), produced in peripheral tissues; inducible Treg cells (ITreg), generated by TGF-beta induction from peripheral naive T cells (*Shevach & Thornton, 2014*). Studies have shown that the prognosis of patients with small cell lung cancer is negatively correlated with FOXP3$^+$CD4$^+$Treg infiltration levels, suggesting that targeting Treg cells is a new therapeutic approach for small cell lung cancer (SCLC) (*Wang et al., 2012*). Other similar studies have shown that an increase in Treg cell content in the peripheral blood and tumors accelerates tumor metastasis and reduces the survival probability of patients (*Petersen et al., 2006*; *Shimizu et al., 2010*; *Erfani et al., 2012*). However, *Ferreira et al. (2020)* found that type 1 Treg cells can enhance the immune barrier in some peripheral tissues, and other studies have reported that some immune cells, such as Tregs, have a bidirectional effect on tumor inhibition and promotion in clinical therapy (*Lakshmi Narendra et al., 2013*). These studies indicate that Tregs play an important role in the progression of LUAD; however, the specific role of Tregs remains

unclear. Therefore, the role of Tregs in immunotherapy in patients with LUAD should be explored.

To fully understand the potential and limitations of targeted Treg cell therapy for cancer, this study explored a LUAD-related prognostic model based on Tregs, to determine the clinical value of targeted Treg cells. To build a gene expression network, this study applied WGCNA algorithm to identify LUAD-related module genes and select brown module genes for in-depth analysis (*Langfelder & Horvath, 2008*). A prognostic model of LUAD related to Treg cells was constructed, and its relationship with the immune microenvironment, chemotherapy, and immunotherapy was analyzed. Representative model genes were verified using western blotting. In this study, the prognostic model was used as a new diagnostic tool to provide more effective treatment guidance for LUAD.

## MATERIALS AND METHODS

### Data acquisition

LUAD was searched in the Gene Expression Omnibus (GEO) database, and the screening criteria were to detect datasets with large genetic data and provide detailed cell type and status information. Finally, five datasets were included, and random allocation was conducted using the randomizr software package in R. These were then combined into two datasets. The test dataset of LUAD is GSE43458, GSE50081 and GSE68465. Two datasets, GSE42127 and GSE72094, are selected as the validation dataset. After removing the samples not belonging to LUAD patients, the final test dataset consisted of 699 samples and 12,477 genes, while the validation dataset included 575 samples and 15,337 genes. Batch calibration and data normalization were performed using the Sav and Limma software packages in R, and batch effect correction was carried out by calling the ComBat function (*Zhang, Parmigiani & Johnson, 2020*; *Ritchie et al., 2015*). When a gene corresponded with multiple probes, the average value was used as the final expression value.

### Treg score calculations for the test dataset samples and evaluation of the relationship between Treg expression values and clinical features

According to the test dataset samples of RNA sequence expression and clinical data, CIBERSORT (https://cibersortx.stanford.edu/) was used to estimate the cancer and the proportion of 22 types of immune cells in the tissue adjacent to the carcinoma; this is a deconvolution algorithm that calculates the gene expression patterns using online tools and provides an estimated abundance of known cell types in a mixed cell population to obtain a Treg value for each sample (*Waks et al., 2019*). Treg expression levels in paired normal and cancerous tissues were compared. In addition, we evaluated the relationship between Treg cell expression and clinical features. To verify the prognostic value of Treg expression, "survival" package of the R software was used to compare overall survival in groups with high and low Treg expression values. The Kaplan–Meier curve was plotted, and the log-rank test was performed according to the corresponding *p*-value to evaluate whether the difference in survival time was significant.

## Immune infiltrating cell estimation and co-expression network construction

Using the expression values of 12,477 genes from the sequencing data, CIBERSORTx was used to estimate the proportion of 22 types of tumor-infiltrating immune cells (*Rusk, 2019*). The R package "WGCNA" was used to construct the weighted co-expression network, which is a biological method for integrating co-expressed genes into the same module, calculating the correlation between the module and sample traits, screening the models with high weights associated with traits, and analyzing the genes in the module to identify target genes (*Langfelder & Horvath, 2008*). The scores of 22 kinds of immune infiltrating cells were taken as sample traits, and the optimal soft threshold power (β) was selected to construct the scale-free network when the scale-free topological index was 0.90. Next, a "dynamic tree cutting" algorithm was used to assign genes with similar expression patterns to the same modules (minimum size = 60). In addition, we estimated the correlation between the module signature genes and the infiltration levels of 22 types of immune infiltrating cells and screened the significance of the modules using Pearson's test. Finally, we selected the "Treg" subtype that we were interested in, and conducted in-depth study on the module genes that were most correlated with Treg.

## Prognostic model building

From the test dataset, 564 samples with complete gene expression profiles and survival data were obtained. LASSO analysis based on the R software package "glmnet" combined with multiple Cox regression analysis was used to screen genes significantly associated with LUAD survival data, and then these genes were applied to construct a prognostic model (*McEligot et al., 2020*). The coefficients calculated by multivariate Cox regression were used in the following formula: risk score = sum of coefficients × prognostic gene expression level (*Lee et al., 2018*). The training set samples were divided into high- and low-risk groups according to the risk scores. The Kaplan–Meier curve, receiver operating characteristic (ROC) curve, and principal component analysis (PCA) were used. PCA and risk score triptychs were used to evaluate the accuracy of the prognosis model.

## Constructing a nomogram

A nomogram was established that combined clinical characteristics such as age, sex, and risk score to calculate the 1-year, 3-year, and 5-year survival of the entire LUAD group. Calibration curves were constructed to evaluate the reliability of the established nomogram (*Hoshino et al., 2018*).

## Pathway differences in high- and low-risk groups identified by GSVA analysis

To further explore the differences in activation status of biological pathways between high- and low-risk groups, "GSVA" R was included in GSVA (*Hänzelmann, Castelo & Guinney, 2013*). The "c2.cp.kegg.v7.4. symbols" geneset from the MSigDB database (https://www.gsea-msigdb.org) were downloaded for GSVA analysis. GSVA is typically used to estimate changes in pathways and bioprocess activities in expression dataset samples using nonparametric and unsupervised methods.

## TME, its immune landscape characterization analysis, and prediction of response to immunotherapy

The Immuno-oncology Biological Research (IOBR) package in R integrates eight commonly used algorithms (MCPcounter, TIMER, IPS, ESTIMATE, xCell, CIBERSORT, EPIC, and quanTiseq) to analyze TME-associated tumor-infiltrating immune cells (TILs) separately using 255 significant gene sets associated with tumor TME, and to evaluate the pattern of immune infiltration in each sample (*Zeng et al., 2021*). The TIL, TME, and immunoinfiltration scores in the high- and low-risk cohorts were calculated, and the scores of each sample were compared to draw conclusions.

## Prediction of chemotherapy response

According to the median risk score, 541 samples were divided into high-risk and low-risk groups, and each group was administered 198 chemotherapy drugs, including camptothecin, axitinib, staurosporin, doramapimod, and ribociclib. The sensitivity of each sample to chemotherapy was predicted using Genomics of Drug Sensitivity in Cancer (GDSC) (http://www.cancerrxgene.org/) (*Cokelaer et al., 2018*). The calculations were performed using the R software package "oncoPredict", and Pearson correlation coefficients were calculated to investigate the correlation between prognostic gene expression and drug therapy sensitivity.

## Western blot analysis

The LUAD cells (A549) and normal human lung epithelial cells (BEAS-2B) were obtained from the Shanghai Institute of Biochemistry and Cell Biology (Shanghai, China) (*Campbell, Main & Fitzgerald, 2013*; *Zhang et al., 2025*; *Schneider et al., 2021*; *Baudat et al., 2025*). Three main surface markers were selected for the detection of Treg cells: FOXP3, CD4, and CD25, which have been defined for the characterization of Treg cells in mammals (*Selvaraj, 2013*).

A549 and BEAS-2B cells were cultured in DMEM containing 10% fetal bovine serum and 1% penicillin-streptomycin at 37 °C and 5% $CO_2$. The logarithmically grown BEAS-2B and A549 cells were added to 100 μL lysate (lysate = RIPA: phosphoprotease inhibitor = 1:100) and cracked on ice for 20 min. The supernatant was collected *via* centrifugation. Using the BCA (catalog no. P0012; Beyotime, Shanghai, China) kit, the protein concentration of the samples was roughly measured. Finally, the loading volume of each sample was uniformly set to eight μL. At the same time, the same amount of proteins were separated by 10% SDS-PAGE gel, and then transferred onto a PVDF membrane. The membrane was washed with TBST buffer and closed with 5% skim milk powder at room temperature for 120 min. Next, the PVDF membrane was incubated with antibodies (1:1,000 dilution) against FOXP3 (Ptgcn, catalog number: 22228-1-AP), SCN9A (Ptgcn, catalog number: 20257-1-AP), CD25 (Ptgcn, catalog number: 30449-1-AP), CD4 (Ptgcn, catalog number: 19068-1-AP), and GAPDH (catalog no. AB0036; Abways, San Diego, CA, USA), and incubated overnight at 4 °C. The following day, the membranes were incubated with a secondary antibody (anti-rabbit immunoglobulin) (Ptgcn, catalog no. SA00001-2) was diluted 1:5,000 and incubated at room temperature for 2 h. The membrane was washed

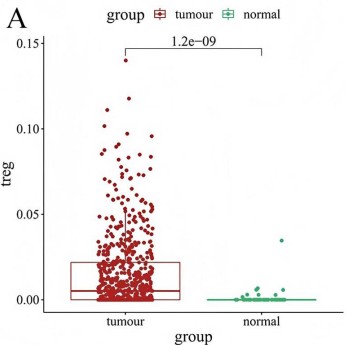 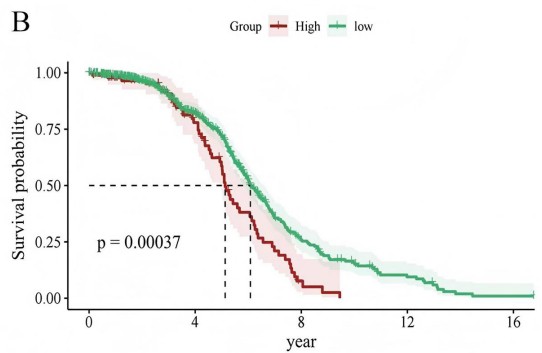

**Figure 1 Relationship between Treg expression value and clinical characteristics of LUAD samples.** (A) Differential expression of Treg expression value between tumor and normal samples; (B) Kaplan–Meier curve of patients with high and low Treg expression value.

again and tested using an Omni-ECL™ Ultra Sensitive Chemical Luminescence Assay Kit (catalog No. SQ201; Epizyme, Cambridge, MA, USA).

## Statistical analysis

In this study, R (version 4.3.3) was used for statistical analysis. The Student's $t$-test was employed to compare the two groups of normally distributed variables, to assess the differences between the groups, and the results were presented as mean $\pm$ standard deviation (SD). For continuous variables, the Wilcoxon test was used. The Kaplan–Meier curve was employed to compare the survival differences among different risk groups, and then the log-rank test was conducted.

# RESULTS

## Treg in LUAD

We used CIBERSORT to evaluate the value of 22 types of immune cells in our analysis, and the results showed that cancer samples had higher Treg expression levels than paracancerous tissues ($P = 1.2e-09$) (Fig. 1A). In addition, patients with LUAD with high Treg expression had a worse prognosis than those with low Treg expression ($P = 0.00037$) (Fig. 1B). These results suggest an association between Treg expression and the clinical characteristics of patients with LUAD.

## Sequencing of gene data of immune infiltrating cells, and construction of co-expression networks

The test dataset of 699 samples (650 tumor samples and 49 normal samples) of 12,477 gene expression data was used to calculate the abundance of 22 types of immune cells using CIBERSORTx. Next, 22 parts of the immune infiltrating cells with gene expression data were selected as WGCNA algorithm traits, and samples with no statistical differences were removed. A co-expression network was constructed for 12,477 gene expression data from 22 types of immune infiltrating cells in LUAD samples. A suitable soft threshold power ($\beta = 2$) was selected so that the scale-free topological index reached a power value of 0.90

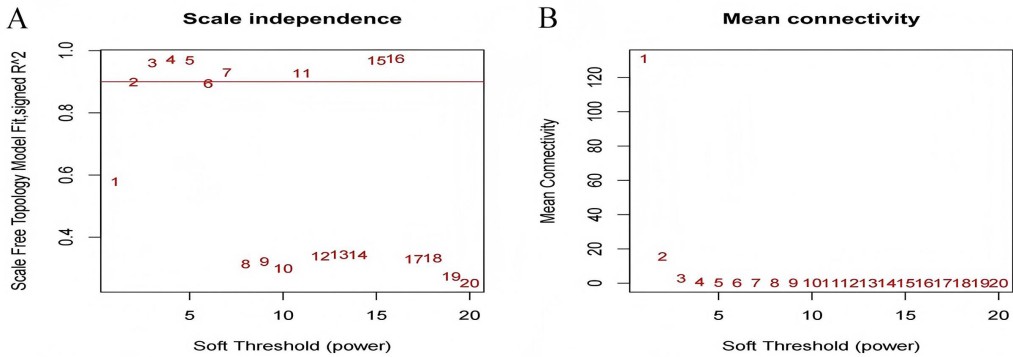

**Figure 2  Select an appropriate soft threshold (power).** (A) The selection of soft threshold makes the index of scale-free topology reach 0.90; (B) The average connectivity analysis of 1–20 soft threshold power.

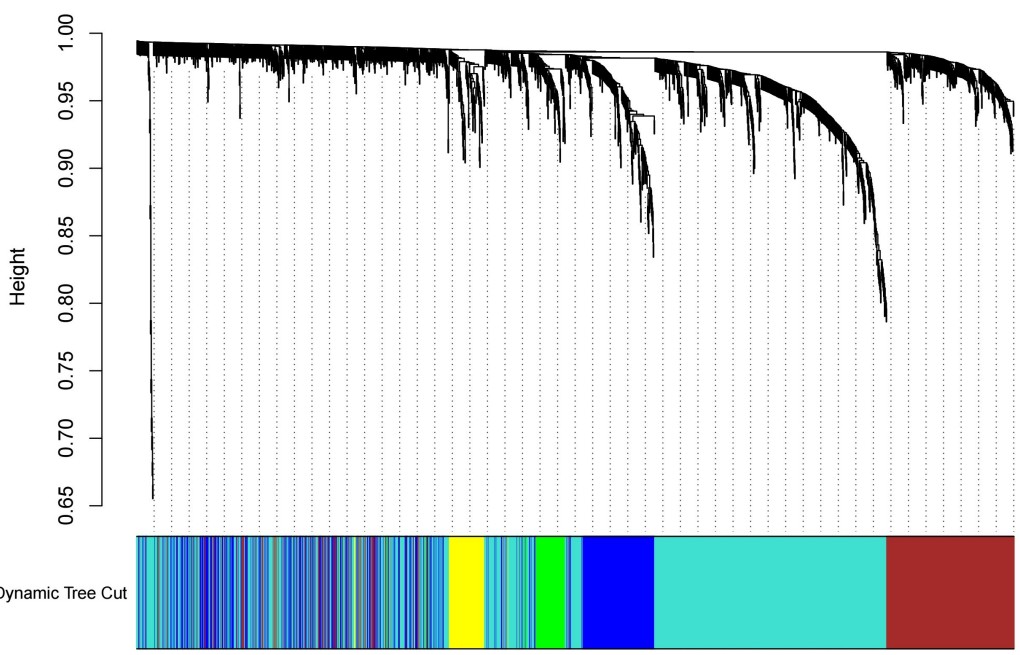

**Figure 3  Build a hierarchical clustering tree.** Uses the dynamic cutting tree algorithm to merge Treg-related genes with similar expression patterns into the same module to form the correlation heat map of immune infiltrating cells (traits) of the characteristic genes of the hierarchical cluster tree.

(Figs. 2A, 2B). Treg-related genes with similar expression patterns were merged into the same module using a dynamic cut-tree algorithm (module size $= 60$) to form a hierarchical clustering tree with five modules (Fig. 3). As shown in Fig. 4, among the five modules, 405 genes in the brown module were highly correlated with Tregs ($R^2 = 0.43$, P $= 1e-30$). Subsequently, brown module genes were selected to construct a prognostic model (*Liu et al., 2021*).

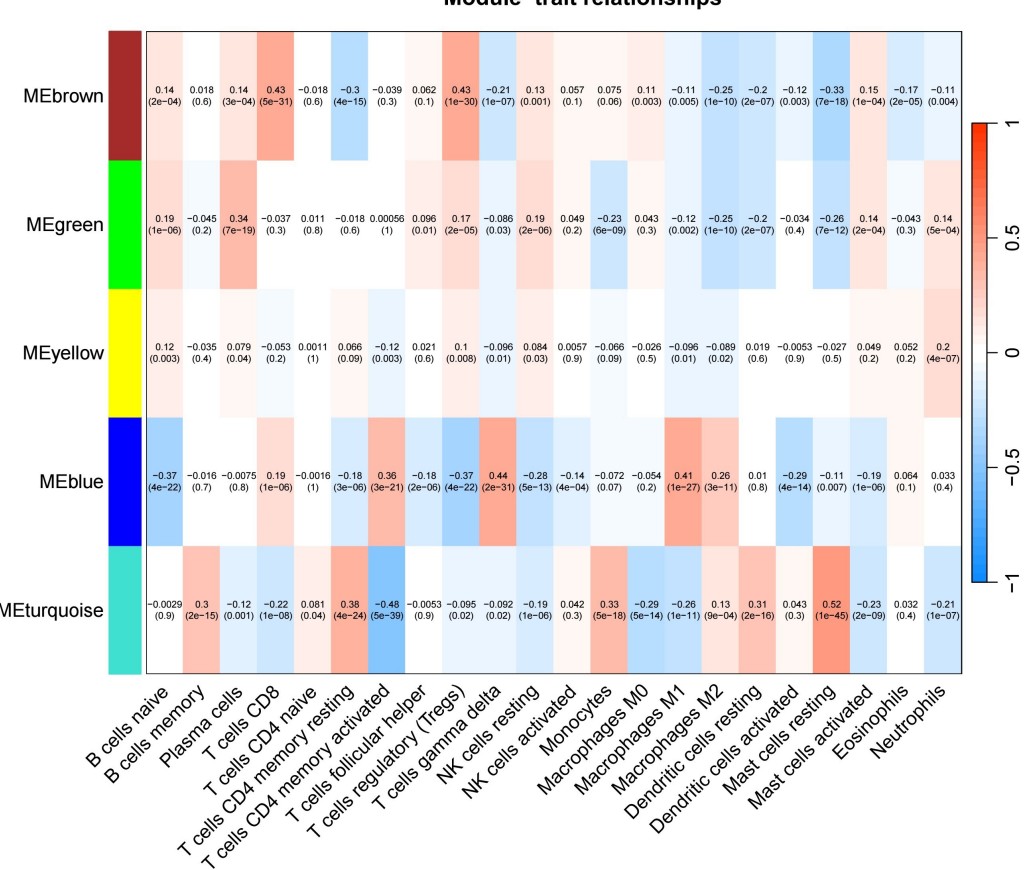

**Figure 4 Generate a correlation heatmap between each module and immune infiltrating cells (characteristics).** Correlation heat maps of immunoinfiltrating cells (traits) of module characteristic genes In each box, the top digit is the coefficient between the cell infiltration level and the corresponding module, and the bottom digit is the *p* value.

## Construction and validation of key gene identification and prognostic model

To evaluate the prognostic performance of the brown module genes, a univariate Cox regression analysis was performed on 405 genes (Table S1, Fig. S1C). In the test dataset, 16 genes were strongly associated with OS in patients with LUAD ($p < 0.001$). These 16 genes were analyzed by LASSO regression, and 12 genes were screened (Figs. S1A, S1B) for multivariate Cox regression analysis, of which six genes were significantly associated with the risk ratio in patients with LUAD (Fig. S1D). These genes have been previously used to develop prognostic models. The six genes were weighted by relative coefficients, The formula was: risk score = (0.1688*ADARB2) + (0.1802*B3GALT1) + (−0.1363*FER) + (0.2407*LTB4R2) + (−0.1207*N6AMT1) + (−0.0899*SCN9A). ADARB2 (HR = 1.1838 (1.0275–1.3639), $P = 0.0195$), B3GALT1 (HR = 1.1974 (1.0331–1.3879), $P = 0.0167$), FER (HR = 0.8726 (0.7763–0.9808), $P = 0.0223$), LTB4R2 (HR = 1.2722 (1.1052–1.4643), $P = 0.0008$), N6AMT1 (HR = 0.8863 (0.8059–0.9747), $P = 0.0128$), SCN9A (HR = 0.9140

**Table 1  Multivariate cox regression weighted analysis of six genes according to their relative coefficients.**

| id | coef | HR | HR.95L | HR.95H | pvalue |
|---|---|---|---|---|---|
| ADARB2 | 0.1688 | 1.1838 | 1.0275 | 1.3639 | 0.0195 |
| B3GALT1 | 0.1802 | 1.1974 | 1.0331 | 1.3879 | 0.0167 |
| FER | −0.1363 | 0.8726 | 0.7763 | 0.9808 | 0.0223 |
| LTB4R2 | 0.2407 | 1.2722 | 1.1052 | 1.4643 | 0.0008 |
| N6AMT1 | −0.1207 | 0.8863 | 0.8059 | 0.9747 | 0.0128 |
| SCN9A | −0.0899 | 0.914 | 0.841 | 0.9933 | 0.0341 |

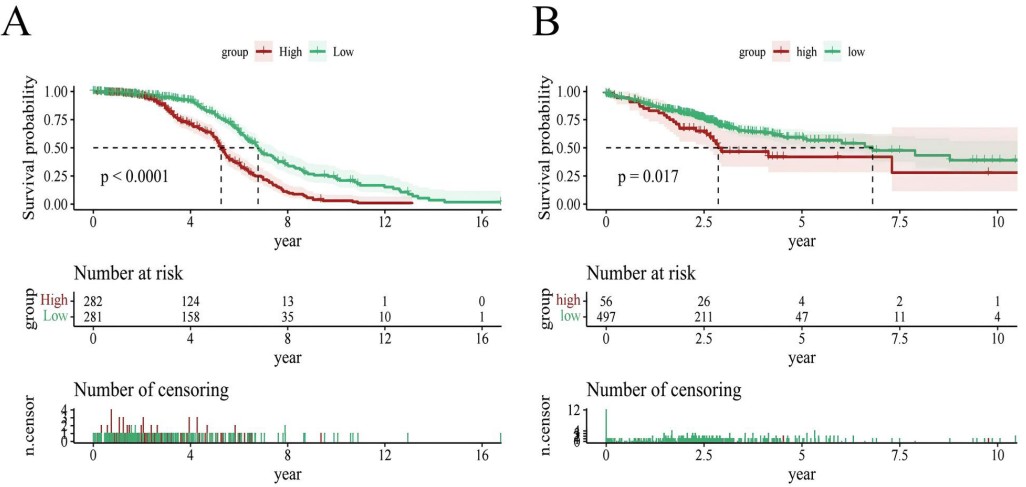

**Figure 5  Survival analysis.** (A) Survival analysis plot of the modeling dataset grouped by high and low risk scores; (B) Survival analysis plot of the validation dataset grouped by high and low risk scores.

(0.8410–0.9933), $P = 0.0341$) (Table 1). The risk scores of the samples in the test dataset were calculated according to the above formula and divided into high- and low-risk groups. There was a statistically significant difference in overall survival among the groups ($p < 0.0001$, log-rank test) (Fig. 5A). The area under the curve (AUC) for overall survival was 0.68 at 6 years, 0.64 at 3 years, and 0.51 at 1 year (Fig. 6A). PCA revealed significant differences between low- and high-risk patients (Fig. 7A). We ranked the patients' risk scores, and Fig. 8 reflects the life status, survival status distribution, and gene expression patterns of patients with LUAD. To explore the predictive power of the prognostic model, we constructed a complete validation dataset and calculated the risk score for the entire set using the prognostic model calculation formula. According to the risk score, the whole group was divided into high- and low-risk groups, and there was a statistically significant difference in the Kaplan–Meier survival curves between the two groups ($P = 0.017$) (Fig. 5B). In the entire set, the AUC at 1 year was 0.53 and the AUC at 3 and 6 years were 0.56 (Fig. 6B). PCA also showed differences among the groups (Fig. 7B). The risk score, survival status, and expression of the six prognostic genes of the whole group are shown in Fig. 9.

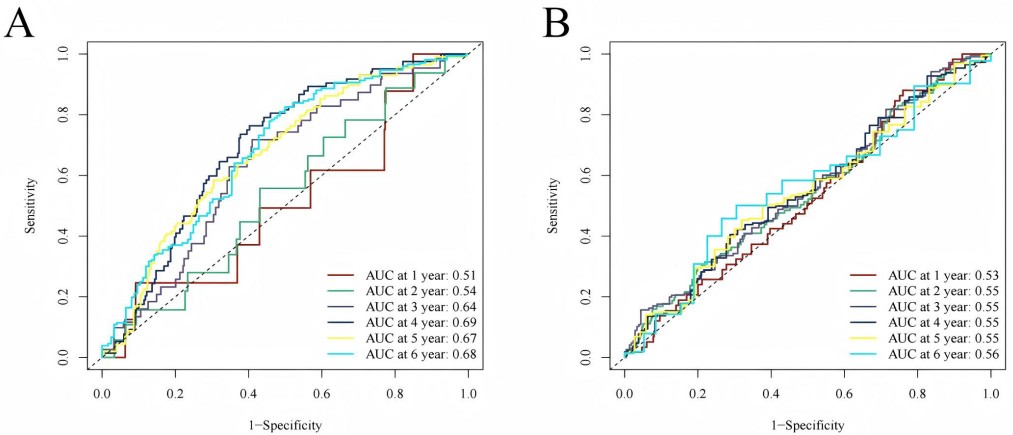

**Figure 6 Time-dependent ROC curve analysis.** (A) Time-dependent ROC curve analysis of the modeling dataset grouped by high and low risk scores; (B) Time-dependent ROC curve analysis of the validation dataset grouped by high and low risk scores.

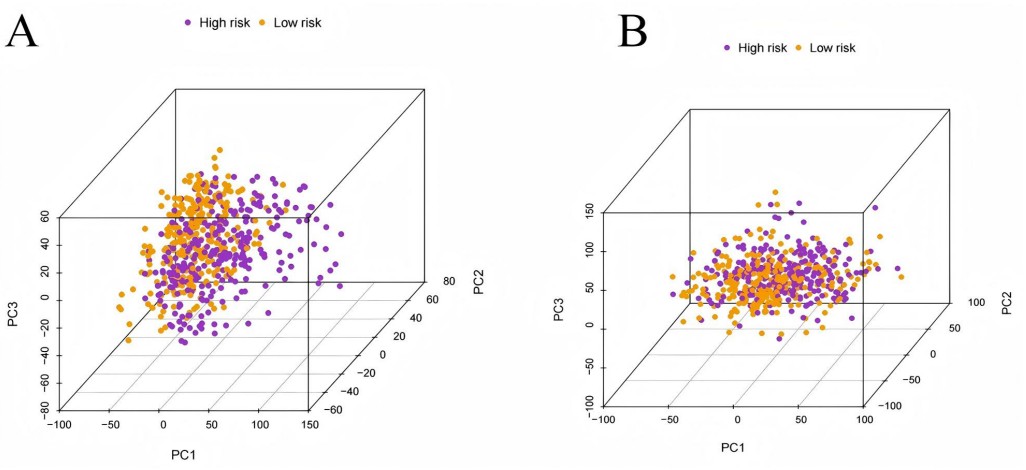

**Figure 7 Principal component analysis.** (A) The principal component analysis graph of the modeling dataset grouped according to high and low risk scores; (B) The principal component analysis graph of the validation dataset grouped according to high and low risk scores.

## Prediction of survival by nomogram

Using known risk scores and clinical features, multivariate logistic regression was used to construct nomograms that could accurately predict patient survival. Age, sex, and risk scores were considered predictors of survival and were incorporated into the nomogram (Fig. 10A). It can be seen that the risk score is the most influential factor in the overall nomogram score. According to the nomogram correction curve (Fig. 10B), the survival rate predicted using the prognostic model was consistent with the observed survival rate. This indicated that the prognostic model has good clinical practicability.

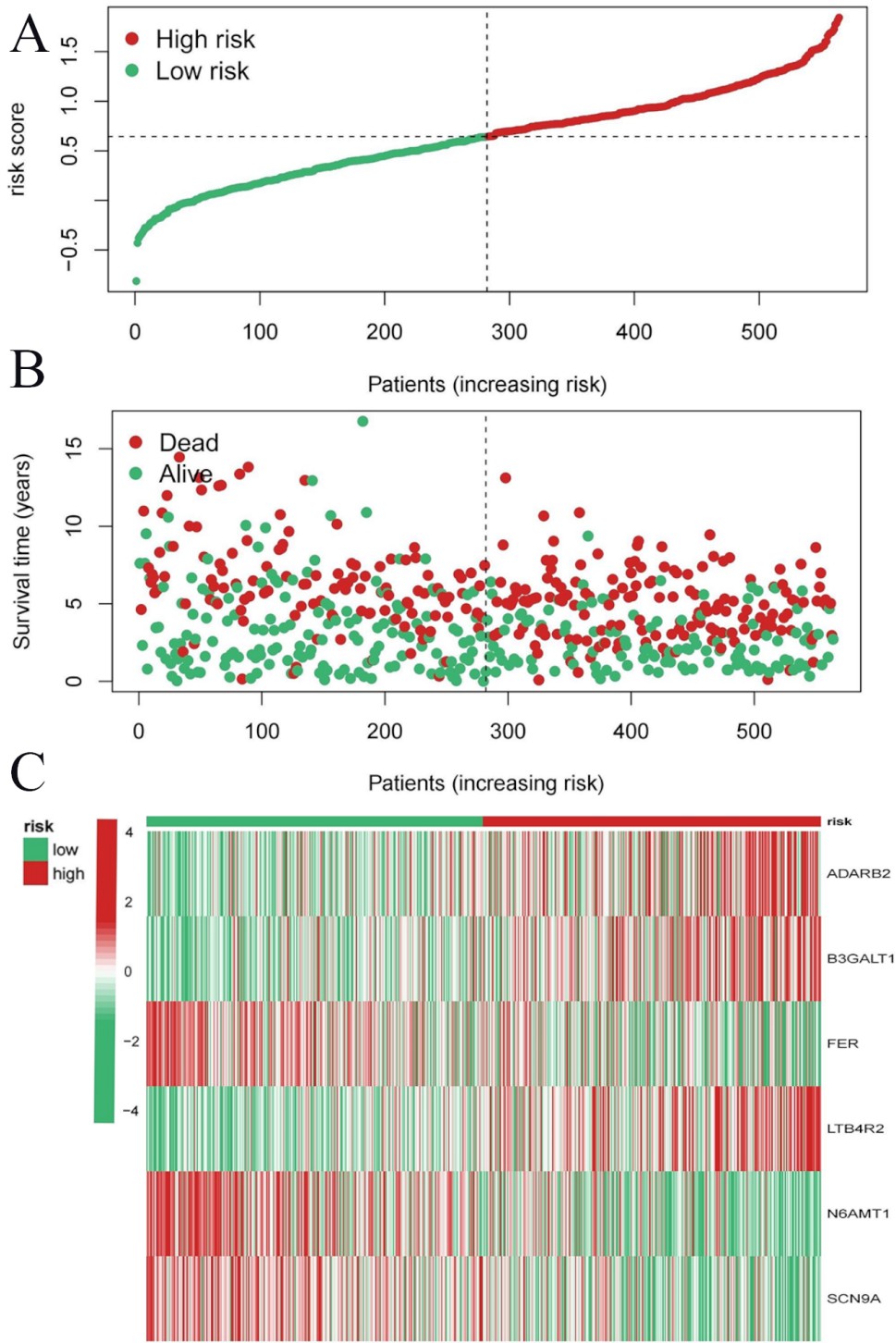

**Figure 8** **Risk triplot of the test dataset.** (A) Risk score plot, with risk scores arranged in ascending order. (B) Scatter plot of survival time, divided into red and green based on survival status, where red represents the death state and green represents the survival state. (C) Heatmap of modeled gene expression levels.

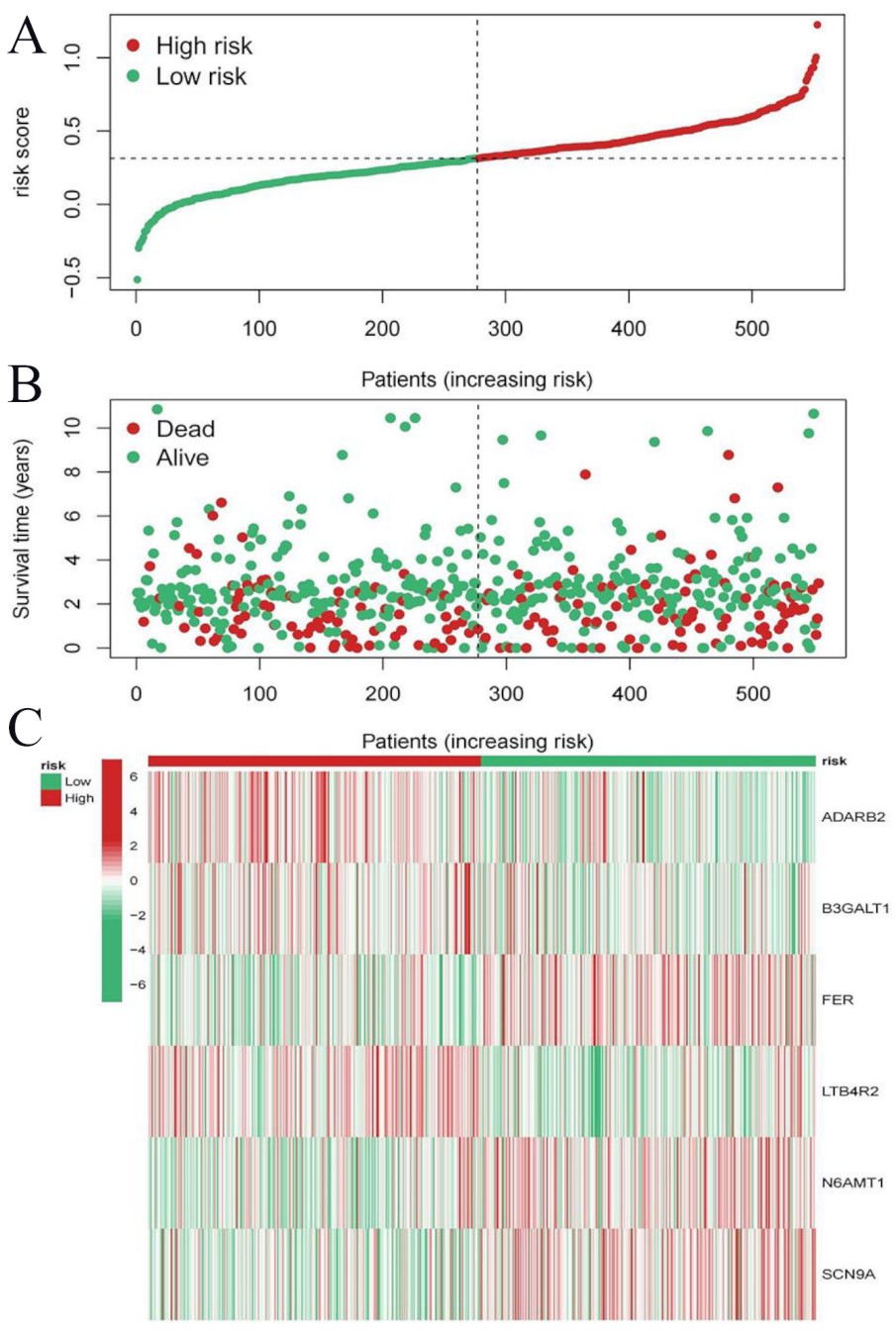

**Figure 9** **Risk triplot of the validation dataset.** (A) Risk score plot, with risk scores arranged in ascending order. (B) Scatter plot of survival time, divided into red and green based on survival status, where red represents the death state and green represents the survival state. (C) Heatmap of modeled gene expression levels.

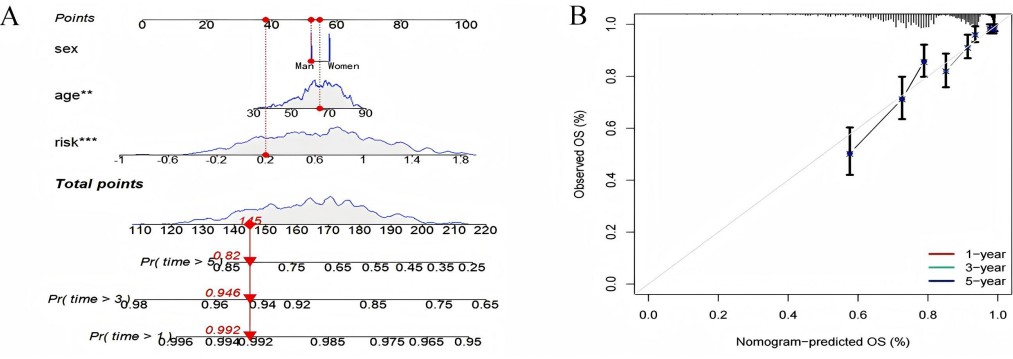

**Figure 10** **Nomogram analysis and calibration curve.** (A) Nomograph (B) Calibration curve to compare the agreement between predicted survival and actual survival (*$p < 0.05$, **$p < 0.01$, ***$p < 0.0001$).

## Pathway differences between high- and low-risk groups identified by GSVA analysis

To explore the differences in the activation status of biological pathways between the high- and low-risk groups, samples were divided into average value according to risk score expression values, and GSVA was performed. A heat map (Fig. 11) was used to visualize the relevant biological processes and arrange them according to the *p*-value. The results showed that carcinogenic activation pathway of the P53 signaling pathway and cell cycle were significantly enriched in the high-risk group. These immune-related pathways regulate tumorigenesis and development, providing a theoretical basis for poor prognosis in high-risk groups. Figure 12 shows the classification of 108 pathways with significant differences, among which the pathway related to the human metabolic system accounted for the most, and the immune-related pathway accounted for 19.23% of the differences.

## Risk score associated with TME signaling and immune cell infiltration

To further explore whether risk scores were associated with the TME in LUAD, we used the IOBR to define TME signaling. Tumor-related signal scores were higher in high-risk group than in the low-risk group. These signals included genes involved in cell cycle regulation, DNA damage repair, mismatch repair and homologous recombination, molecular carcinoma m6A, exosomes, positive regulation of exosome secretion, and ferritin deposition (Figs. 13A, 14). Next, we used eight typical algorithms (MCPcounter, TIMER, IPS, ESTIMATE, xCell, CIBERSORT, EPIC, and quanTiSeq) to estimate T cell infiltration, which showed lower scores for low-risk group (Fig. 15). Contrary to the T-cell invasion results, most neutrophils showed more infiltration in the low-risk group; however, this difference was generally less pronounced (Fig. 16). However, in natural killer (NK) cells, differences were widespread, and scores were higher in the high-risk group (Fig. 13B). Based on the results of the infiltration of different immune cells using various algorithms, it was concluded that the TME had a stronger immunosuppressive effect in the high-risk group.
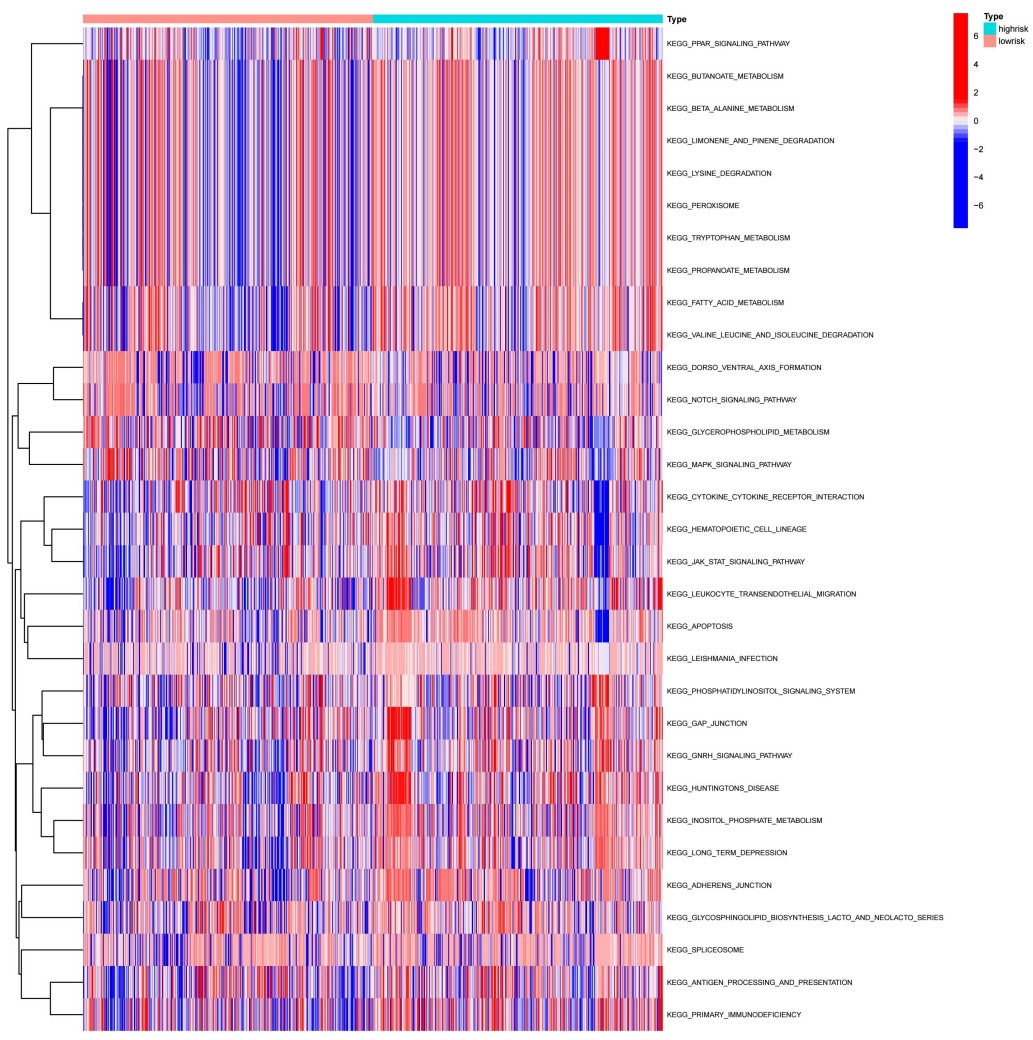

**Figure 11** **Heatmap of high-risk groups in GSVA.** Red represents activated pathways, while blue represents inhibited pathways.

## Relationship between prognostic model and chemotherapy sensitivity

Chemotherapy is a conventional treatment for LUAD. We predicted and analyzed the responses to 198 chemotherapeutic drugs, including camptothecin, axitinib, staurosporin, *etc.* For patients in the high- and low-risk groups, all drugs showed significant differences between groups (Fig. 17). Figures 18A, 19A, and 20A show that paclitaxel ($R^2 = 0.25$, $P = 7.7e{-}11$), rapamycin ($R = 0.28$, $P = 2e{-}13$), and staurosporin ($R = 0.19$, $P = 1.1e{-}06$) positively correlated with the risk score. The box plot (Figs. 18B, 19B, and 20B) showed higher sensitivity to paclitaxel ($P = 2.9e{-}05$), rapamycin ($P = 2.4e{-}07$), and staurosporin ($P = 0.00052$) in the low-risk group, all of which suggest that chemotherapy agents may have greater efficacy in the low-risk group.

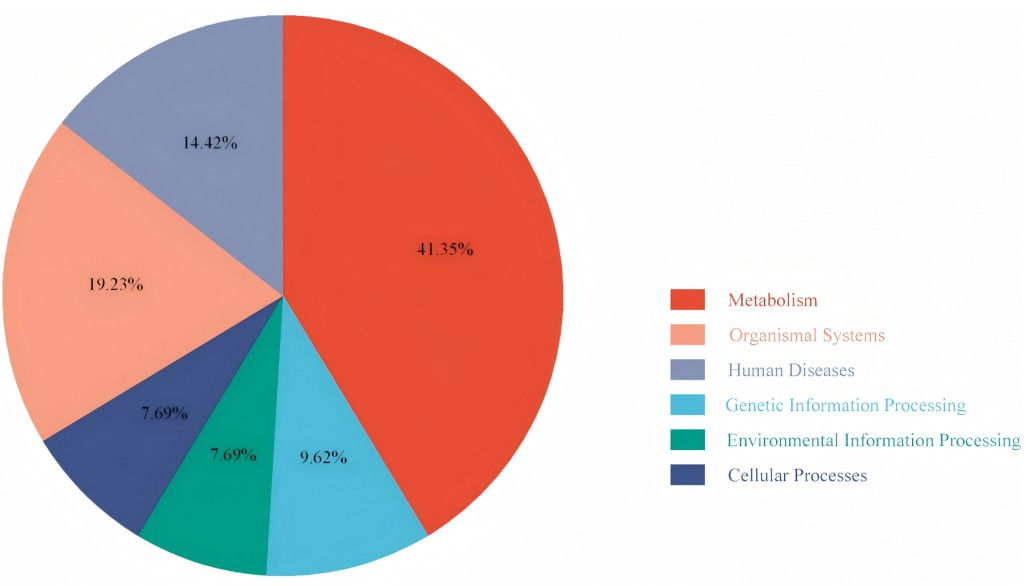

**Figure 12  Pie chart of the GSVA results.** It shows the proportion of different pathways.

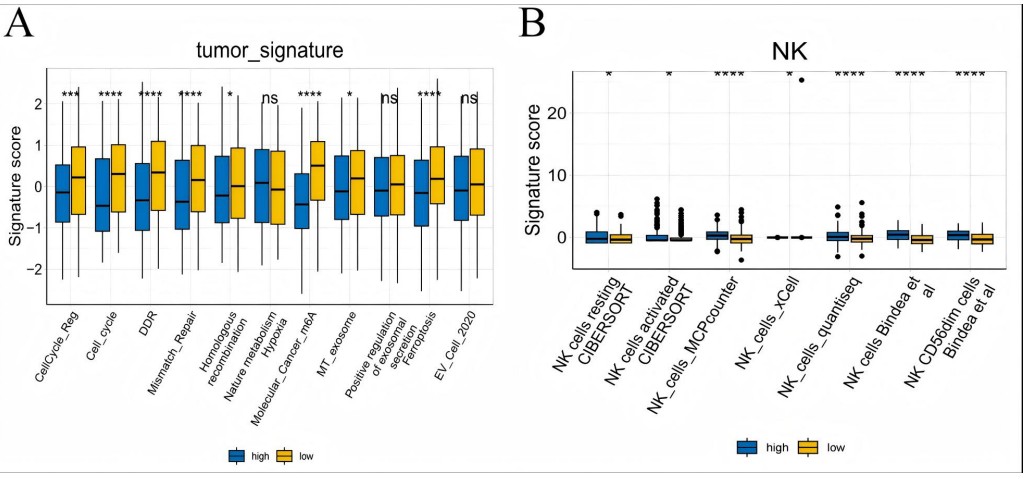

**Figure 13  Box plots of tumor-related features and NK cell analysis of tumor-infiltrating immune cells in the IOBR analysis of the prognostic model and immune relationship.** (A) Box plot of tumor-related features; (B) NK cell analysis of tumor-infiltrating immune cells in the prediction algorithm ($*p < 0.05$, $**p < 0.01$, $***p < 0.0001$).

## Western blot analysis verified the expression of prognostic model genes in LUAD

In the validation dataset, *SCN9A* was found to have the highest negative correlation with the risk score (R = −0.54, $p < 2.2e{-}16$) (Figs. 21A, 21B, 21C, 21D, 21E, and 21F). Subsequently, the proteins of A549 and BEAS-2B cells were collected for western blot analysis of the modeling gene *SCN9A* and Treg marker genes FOXP3, CD4, and CD25. The
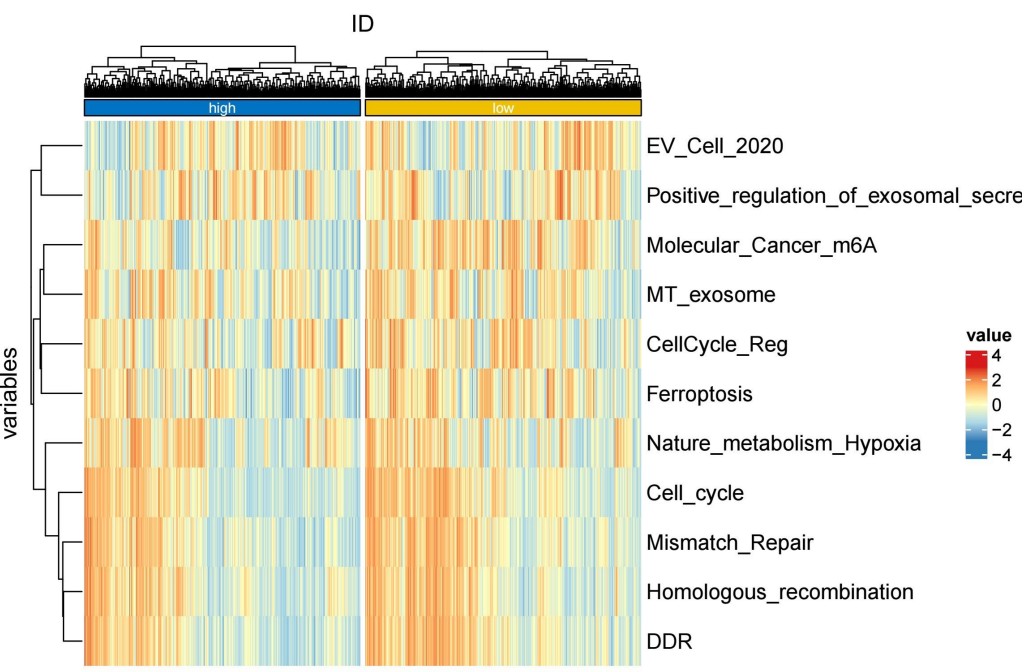

**Figure 14  Heatmap of tumor-related features in the IOBR analysis of the prognostic model and immune relationship.**

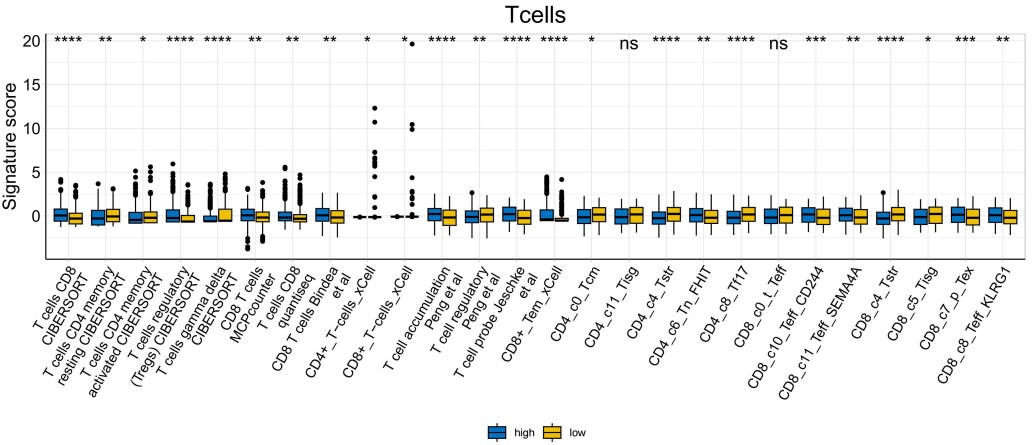

**Figure 15  T cell analysis of tumor-infiltrating immune cells in the IOBR analysis of the prognostic model and immune relationship (\*$p < 0.05$, \*\*$p < 0.01$, \*\*\*$p < 0.0001$).**

results of western blot analysis showed that the expression of Treg marker genes FOXP3, CD4, and CD25 in LUAD cells was upregulated compared with that in the BEAS-2B cells of normal lung tissue, whereas the expression of *SCN9A* was downregulated (Figs. 21G, 21H, and 21I). This is consistent with the results of previous dataset analysis. Treg expression was increased in LUAD compared with that in normal lung tissue, and the *SCN9A* was

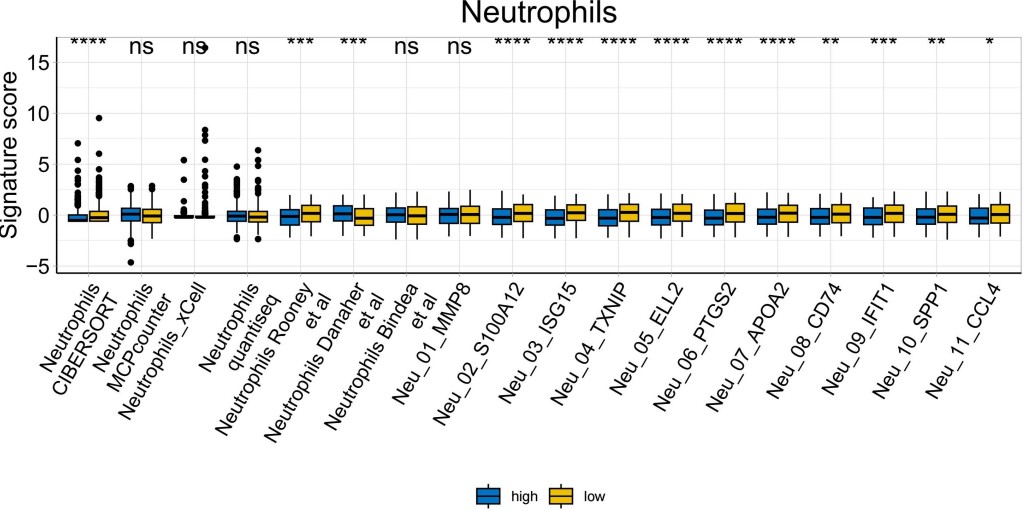

**Figure 16** Neutrophil analysis of tumor-infiltrating immune cells in the IOBR analysis of the prognostic model and immune relationship (*$p < 0.05$, **$p < 0.01$, ***$p < 0.0001$).

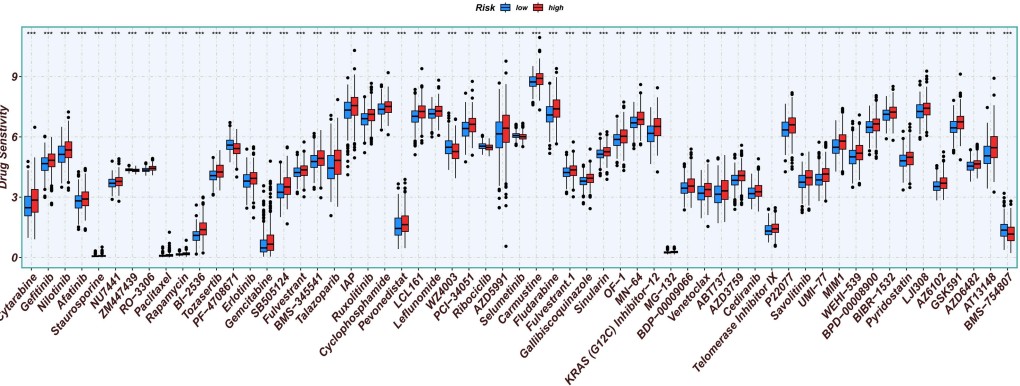

**Figure 17** **Analysis results of risk score and chemotherapy sensitivity.** Relationship between high- and low-risk group and chemotherapy sensitivity. (*$p < 0.05$, **$p < 0.01$, ***$p < 0.0001$).

negatively correlated in the prognostic model; therefore, its expression was decreased in LUAD.

## DISCUSSION

As one of the main histological phenotypes of lung cancer, LUAD has poor prognosis (*Travis et al., 2015*; *Shen et al., 2023*). Research on tumor immune microenvironment and the rapid development of immunotherapy have greatly improved the prognosis and survival of patients with LUAD. However, there are still drawbacks of immunotherapy for LUAD, such as increased drug resistance and frequent adverse reactions (*Liu et al., 2022*). Therefore, it is crucial to explore potential safe and effective immune-related targets to improve LUAD immunotherapy outcomes (*Tsimberidou et al., 2020*). Treg cells, an

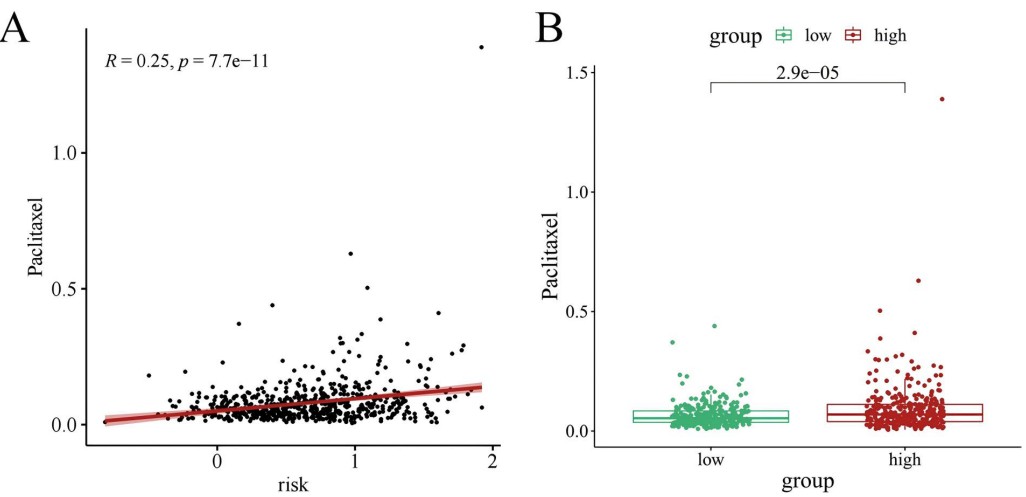

**Figure 18** **The drug sensitivity analysis results of paclitaxel.** (A) Scattered correlation coefficients between paclitaxel and risk score. (B) Box chart of difference between paclitaxel and high- and low-risk group.

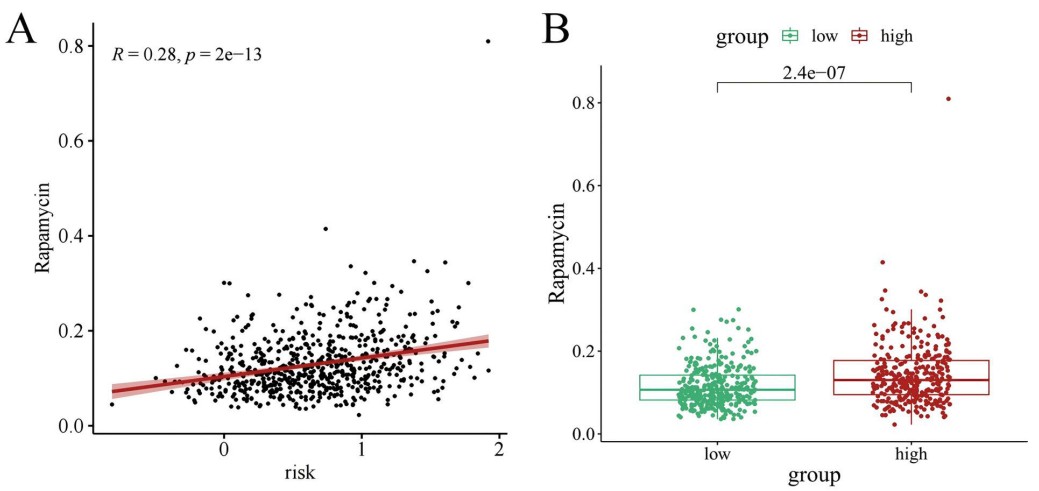

**Figure 19** **The drug sensitivity analysis results of rapamycin.** (A) Scattered correlation coefficients between rapamycin and risk score; (B) Box chart of difference between rapamycin and high- and low-risk group.

important component of the tumor immune microenvironment, are of great significance for LUAD immunotherapy. Through single-cell studies, it was found that the number and inhibitory function of Tregs in tumor samples were significantly higher than those in blood, lung, and lymph node samples (*Ferreira et al., 2020*). *Akimova et al. (2017)* also showed that the higher the proportion of tumor-infiltrating Tregs, the lower the overall survival of patient (*Maruyama et al., 2011*). *Xie, Wei & Xu (2020)* found that Treg cells would assist cancer cells in evading the surveillance of the immune system. Therefore,

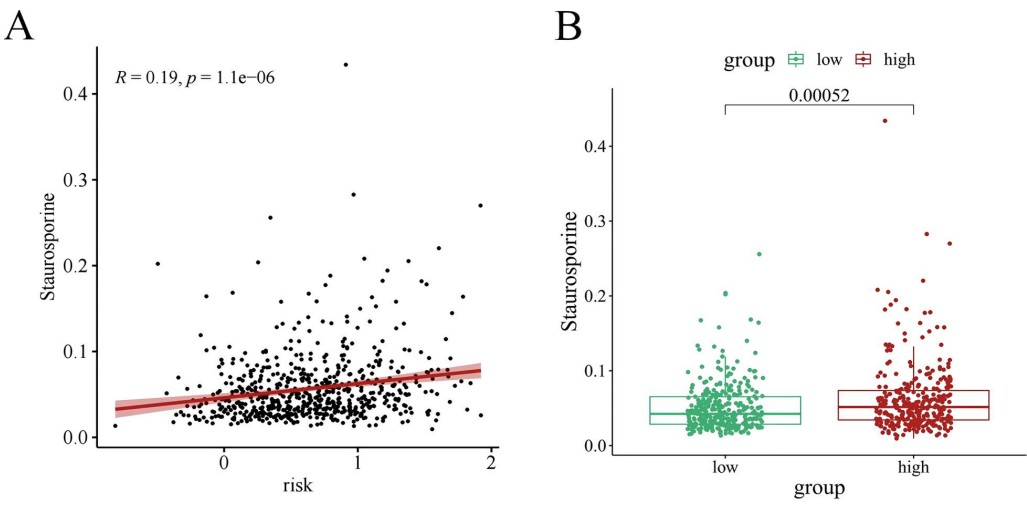

**Figure 20** **The drug sensitivity analysis results of staurosporin.** (A) Scattered correlation coefficients between staurosporin and risk score; (B) Box chart of the difference between staurosporin and high risk groups (*$p < 0.05$, **$p < 0.01$, ***$p < 0.0001$).

exploring the potential immune-related targets of Treg cells will provide a new perspective for immunotherapy and prognosis of LUAD and solve more practical clinical problems.

In this study, multiple GEO datasets were integrated into test and validation datasets. A prognostic model based on Treg immune cells for LUAD was constructed in the test dataset and validated on the validation dataset. The results demonstrated that this prognostic model can be used to predict the prognosis and immune treatment response of LUAD patients to a certain extent in clinical practice (*Wang et al., 2021*; *Liu et al., 2024*). The CIBERSORT immune infiltration algorithm was first used to evaluate the proportion of 22 types of immune cells, including Treg cells. The results showed that the higher the expression of Treg cells, the lower the overall survival rate of patients. The WGCNA gene co-expression algorithm and correlation analysis were used to identify brown module genes with the highest correlation with Tregs. Univariate Cox, LASSO, and multivariate Cox regression analyses were used to establish a prognostic model, calculate the risk score, and show the prognostic significance of clinical factors. In both the test and validation datasets, AUC and PCA showed similar trends in overall survival, survival status, risk score distribution, gene expression pattern, and risk score in the high- and low-risk group, this indicates that this prognostic model has predictive capabilities. This nomogram showed the highest risk score, indicating that this model could accurately evaluate the prognosis of LUAD (*Kalbasi & Ribas, 2020*). The samples were divided into two groups according to risk score, and the differences in pathway enrichment between the high- and low-risk groups were analyzed. It was found that the oncogenic activation pathways in the high-risk group were significantly enriched in the P53 signaling pathway and the cellcycle. The abnormal activation of these two pathways would exacerbate the immune suppression mediated by Tregs. The inactivation mutation of the key P53 gene would lead to the expansion of Tregs., at the same time, the excessive activation of the cellcycle

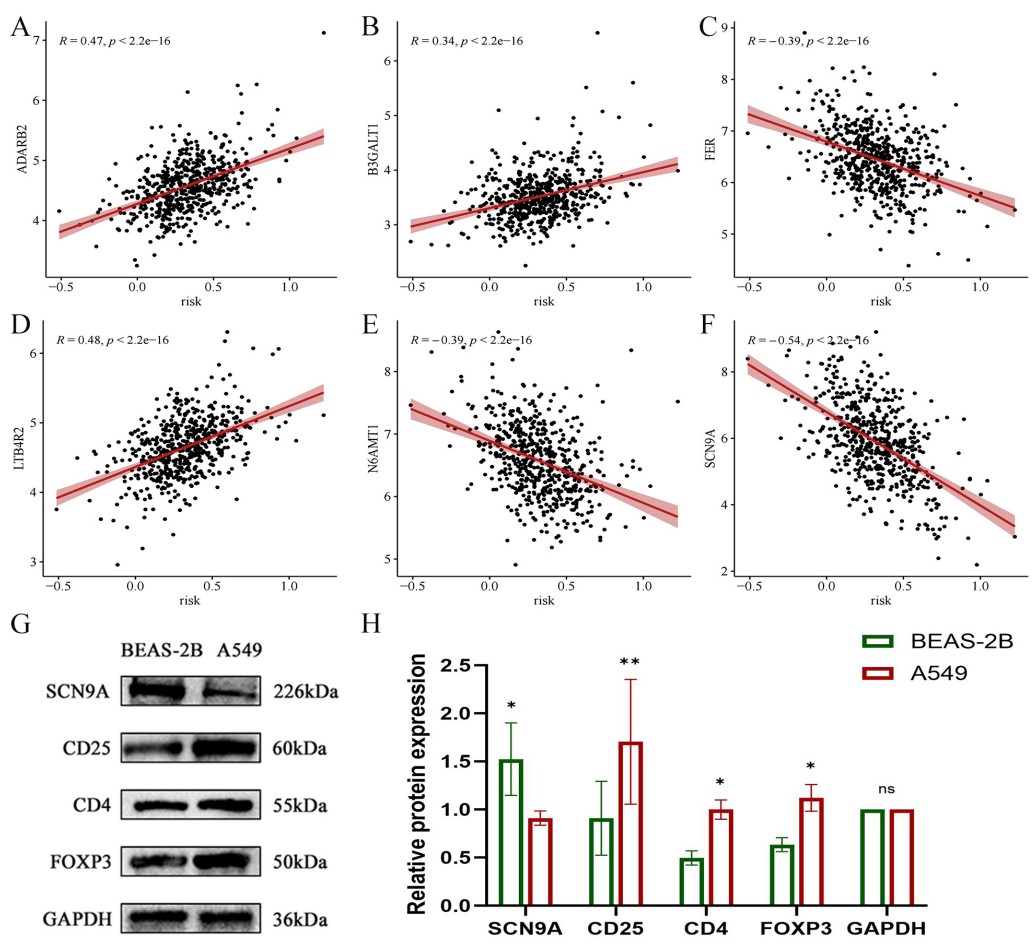

**Figure 21  Scatter plot of correlation coefficient between modeling gene and risk score and Western blot analysis verified.** (A) Scatter plot of correlation coefficient between *ADARB2* and risk score; (B) Scatter plot of correlation coefficient between *B3GALT1* and risk score; (C) Scatter plot of correlation coefficient between *FER* and risk score; (D) Scatter plot of correlation coefficient between *LTB4R2* and risk score; (E) Scatter plot of correlation coefficient between *N6AMT1* and risk score; (F) Scatter diagram of correlation coefficient of *SCN9A* and risk score; (G) Expression difference of each gene in normal lung epithelial cells BEAS-2B and LUAD cell A549; (H) Histogram of expression difference of each gene in normal lung epithelial cells BEAS-2B and LUAD cells A549 (*$p < 0.05$, **$p < 0.01$, ***$p < 0.0001$).

would promote the secretion of immunosuppressive factors (such as IL-10) by Tregs, all of which would accelerate tumor development (*Goel et al., 2017*). Using IOBR to define the TME signals, it was found that the TME has a stronger immunosuppressive effect in the high-risk population, involving processes such as cell cycle regulation, DNA damage repair, molecular cancer m6A, exosome, and other processes. For example, TGF-β and PD-L1 in the cells can be expressed through exosomes, and when they are abnormally activated, they will enhance the immune escape response (*Ye et al., 2021*). In the drug sensitivity analysis, all drugs showed significant differences between the groups, and the low-risk patients had higher sensitivity to star-forming staurosporin (PKC inhibitor), paclitaxel

(microtubule inhibitor), and rapamycin (mTOR inhibitor). These results all indicate the potential therapeutic prospects for risk score stratification in clinical practice.

In-depth analysis of the genes in the prognostic model showed that six genes (*ADARB2, B3GALT, FER, LTB4R2, N6AMT1,* and *SCN9A*) were closely related to cancer, among which *ADARB2, LTB4R2*, and *SCN9A* played important roles in the occurrence and development of lung cancer (*Hua et al., 2022*; *Wu et al., 2022*; *Zeng et al., 2024*; *Sauta et al., 2021*; *Wang et al., 2023*; *Campbell, Main & Fitzgerald, 2013*). Correlation analysis was conducted immediately, and it was found that *SCN9A* had the highest correlation with the risk score and was selected for subsequent western blot analysis. *SCN9A* (sodium channel protein type 9 subunit alpha) is a coding voltage-gated sodium channels of alpha subunit protein family of the first type IX (Nav1.7) gene, and is expressed in the primary sensory neurons and sympathetic ganglion neurons (*Dib-Hajj, Black & Waxman, 2009*). Studies have found that *SCN9A* gene mutations change the structure of the sodium ion channel, affect its normal function, and lead to a variety of diseases related to abnormal pain perception (*Stadler, O'Reilly & Lampert, 2015*; *Suter et al., 2015*; *Cox et al., 2006*). The research has clearly demonstrated that the functional expression of SCN9A is associated with the enhanced invasive ability of NSCLC cell lines, and it is significantly upregulated in tumor tissues. Western blot analysis showed that the expression of Treg cells was verified using the Treg cell characterization genes CD4, CD25, and FOXP3; the verification results showed that the expression of Treg was increased in LUAD, while the expression of the modeling gene *SCN9A* was decreased in LUAD cells.

Combining the western blot analysis results with previous database analysis results, we found that Tregs play an immunosuppressive role in LUAD. In addition, six genes in the prognostic model constructed for Tregs (*ADARB2, B3GALT, FER, LTB4R2, N6AMT1*, and *SCN9A*), particularly SCN9A, are potential therapeutic targets. However, the conclusion of this study is limited to the statistical correlation between gene expression and clinical outcomes. The specific mechanisms of action of genes such as SCN9A in the regulation of Tregs or tumor progression need to be verified through subsequent experiments.

## CONCLUSION

Through a series of bioinformatics and western blot analyses, this study established a prognostic model based on Treg cells, which can be used as a potential target for LUAD immunotherapy. This model was constructed and validated based on retrospective transcriptomics data, and was also verified *in vitro*. However, its clinical efficacy still needs to be verified in a prospective patient cohort.

### Funding

This article was funded by the Shandong Provincial Natural Science Foundation Joint Innovation Fund for Innovation (ZR2023LZY006), the Shandong Provincial Natural Science Foundation Youth Project (Grant No. ZR2023QH095), the Shandong

Provincial Traditional Chinese Medicine Science and Technology Project (Z-2022021), the Youth Science and Technology Talent Support Project of Shandong Province (SDAST2024QTB031), and the Traditional Chinese Medicine Science and Technology Project of Shandong Province (Grant No. Q-2023079). The funders had no role in study design, data collection and analysis, decision to publish, or preparation of the manuscript.

## Grant Disclosures

The following grant information was disclosed by the authors:

Shandong Provincial Natural Science Foundation Joint Innovation Fund for Innovation: ZR2023LZY006.

Shandong Provincial Natural Science Foundation Youth Project: ZR2023QH095.

Shandong Provincial Traditional Chinese Medicine Science and Technology Project: Z-2022021.

Youth Science and Technology Talent Support Project of Shandong Province: SDAST2024QTB031.

Traditional Chinese Medicine Science and Technology Project of Shandong Province: Q-2023079.

## Competing Interests

The authors declare there are no competing interests.

## Author Contributions

- Tian Zhao conceived and designed the experiments, performed the experiments, analyzed the data, prepared figures and/or tables, authored or reviewed drafts of the article, and approved the final draft.
- Yan Yao conceived and designed the experiments, prepared figures and/or tables, and approved the final draft.
- Yan Sun conceived and designed the experiments, prepared figures and/or tables, and approved the final draft.
- Qingliang Lv analyzed the data, prepared figures and/or tables, and approved the final draft.
- Changgang Sun analyzed the data, prepared figures and/or tables, and approved the final draft.
- Yining Cheng conceived and designed the experiments, performed the experiments, analyzed the data, authored or reviewed drafts of the article, and approved the final draft.
- Chundi Gao conceived and designed the experiments, performed the experiments, analyzed the data, authored or reviewed drafts of the article, and approved the final draft.
- Jing Zhuang conceived and designed the experiments, performed the experiments, analyzed the data, authored or reviewed drafts of the article, and approved the final draft.

## Data Availability

The code and data are available in the Supplemental Files.
The LUAD datasets are available at NCBI:

GSE43458

GSE50081

GSE68465

GSE42127

GSE72094

## Supplemental Information

Supplemental information for this article can be found online at http://dx.doi.org/10.7717/peerj.20287#supplemental-information.

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
