# Peer review of "Integrated bioinformatics screening and experimental validation: construction of a LUAD prediction model based on Treg-related genes"

_PeerJ, doi:10.7717/peerj.20287_

## Round 0.1 · original submission · Major Revisions

· Academic Editor

Major Revisions

Reviewer 1 ·

Basic reporting

.

Experimental design

.

Validity of the findings

.

Additional comments

This study aims to construct and validate a prognostic model for lung adenocarcinoma (LUAD) based on regulatory T cell (Treg)-related genes using integrated bioinformatics and experimental approaches. The authors analyzed LUAD datasets from the GEO database, combining CIBERSORT and weighted gene co-expression network analysis (WGCNA) to identify Treg-associated modules. The brown module, highly correlated with Tregs, was selected for further analysis. Six genes (ADARB2, B3GALT1, FER, LTB4R2, N6AMT1, and SCN9A) were identified via LASSO and Cox regression to build a prognostic model. The model stratified patients into high- and low-risk groups, with the latter showing significantly better survival outcomes. Experimental validation using Western blotting confirmed decreased expression of SCN9A and increased Treg markers (FOXP3, CD4, CD25) in LUAD cells compared to normal lung epithelial cells. The study also highlighted pathway enrichment (e.g., P53 signaling, cell cycle) in high-risk groups and identified differential chemotherapy sensitivities. The prognostic model, supported by nomograms and immune microenvironment analysis, demonstrates potential for clinical application in predicting LUAD outcomes and guiding immunotherapy. SCN9A emerges as a promising therapeutic target. However, issues such as formatting errors in citations and unclear figure accessibility require attention.

1. in the data acquisition section, gse50081 has a total of 181 samples, but in this article, only 127 patients with stage I were used. Similarly, gse42127 has a total of 176 samples, but only 133 of them were used in this article. It is suggested to supplement the reasons for the selection of samples and the impact on the results, so as to enhance the persuasiveness of the sample results.
2. in the part of Western blot analysis, A549 and BEAS-2B cell lines were used for corresponding experiments, but the reasons for using these two cell lines were not explained. It is suggested to supplement the corresponding cell line validation experiment, so as to prove that the expression effect of the target gene is the best in A549 and BEAS-2B cell lines.
3. in the statistical analysis section, it does not explain how to deal with non normal distribution data, such as whether to carry out data conversion. It is suggested to clearly mark the data distribution test method and supplement the applicable conditions of non parametric test.
4. there are many detailed errors in this article. For example, "error! Reference source not found." appears many times in lines 79, 83 and 87 of the background part, and the document number is repeated, such as [14]. It is recommended to carefully check and modify.
5. In the discussion, the authors should discuss and cite the follows:
Wang X, Xiao Z, Gong J, Liu Z, Zhang M, Zhang Z. A prognostic nomogram for lung adenocarcinoma based on immune-infiltrating Treg-related genes: from bench to bedside. Transl Lung Cancer Res. 2021 Jan;10(1):167-182. doi: 10.21037/tlcr-20-822. PMID: 33569302; PMCID: PMC7867791.
Liu Y, Li Z, Meng Q, Ning A, Zhou S, Li S, Tao X, Wu Y, Chen Q, Tian T, Zhang L, Cui J, Mao L, Chu M. Identification of the consistently differential expressed hub mRNAs and proteins in lung adenocarcinoma and construction of the prognostic signature: a multidimensional analysis. Int J Surg. 2024 Feb 1;110(2):1052-1067. doi: 10.1097/JS9.0000000000000943. PMID: 38016140; PMCID: PMC10871637.

Reviewer 2 ·

Basic reporting

1. Important literature is not cited. For instance, a related study (PMID: 37671160) has previously reported on a Treg-based prognostic model in LUAD. This work should be referenced and compared to clarify the novelty and differences of the current model.

2. The overall image quality is poor. Many panels are stretched or pixelated, making interpretation difficult. For example, Figure 5A is unreadable and should be replaced with a higher-resolution, properly formatted version. All figures must be clearly labeled and legible when zoomed.

3. The discussion section is superficial and does not adequately interpret the findings. Important results, such as pathway enrichment (beyond P53 and cell cycle), immune cell differences, and drug sensitivity findings, are presented but not discussed in detail.

Suggestions:
1. Improve figure quality and clarity.

2. Reference and compare with relevant prior studies to establish novelty.

3. Substantially expand the discussion, especially on biological interpretation and translational implications of findings.

Experimental design

1. The authors merged three GEO datasets (GSE43458, GSE50081, GSE68465) into the training set and two (GSE42127 and GSE72094) into the validation set without sufficient rationale or methodological transparency. No batch effect correction results (e.g., PCA or ComBat output) are shown to justify effective integration of datasets.

2. Surprisingly, the widely used TCGA-LUAD dataset was excluded without explanation. This omission should be addressed.

3. Figure 1 only presents the prognostic value of Treg expression in the training set. The same analysis should be conducted in the validation set to assess robustness.

4. In Figure 2A, the choice of a soft threshold of 2 is not well-supported; the plot indicates instability. The authors should justify this choice more clearly.

5. The construction of the prognostic model based on Treg-associated genes is reasonable; however, its verification using the A549 tumor cell line (rather than Tregs or co-culture systems) is questionable given the immunological focus of the model.

Suggestions:
1. Include and justify batch effect correction methodology.

2. Analyzing the TCGA-LUAD dataset.

3. Validate initial Treg findings in an independent dataset.

4. Provide rationale for key parameter choices in WGCNA.

5. Reconsider or justify the use of A549 cells for experimental validation of immune-related genes.

Validity of the findings

1. The model shows moderate to weak prognostic performance. The AUC values for 1-year and 3-year survival are all below 0.6 in both training and validation sets, indicating limited predictive utility. This limitation must be discussed.

2. The authors should consider optimizing the model—e.g., testing different WGCNA thresholds or alternative modeling methods (e.g., Random Forest, XGBoost).

3. Claims that the model offers strong clinical value or identifies therapeutic targets (e.g., SCN9A) are premature. Correlation does not imply causation, and mechanistic or functional experiments are lacking.

4. The biological implications of immune infiltration and pathway enrichment differences are not sufficiently explored in the discussion.

Suggestions:
1. Acknowledge limitations of predictive power.

2. Explore model optimization strategies.

3. Refrain from overstating translational conclusions without functional validation.

4. Deepen interpretation of immunological and pharmacological findings.

Reviewer 3 ·

Basic reporting

This manuscript, "Integrated bioinformatics screening and experimental validation: Construction of a LUAD prediction model based on Treg-related genes," addresses the clinically relevant issue of prognosis in Lung Adenocarcinoma (LUAD) by investigating the role of regulatory T cells (Tregs). The authors commendably combine a robust bioinformatics pipeline—using public datasets to identify Treg-related genes, build a prognostic model, and examine its implications—with essential in vitro experimental validation of a key gene, SCN9A. The overall approach is coherent, and the study's objective is important.

However, the manuscript requires revisions before it can be considered for publication.

The manuscript needs language editing. The text has some ungrammatical and awkwardly phrased phrases, which make it hard for an international audience to understand. For example, in the abstract, "Differences in the expression of the prognostic models were analyzed..." can be rephrased to "The expression of genes within the prognostic model...". Similar issues are found throughout the manuscript.

The figures are generally relevant and of acceptable quality. However, the quality could be improved, such as those in Figure 6, with label sizing and the quality of some panels, like C and D, needing adjustment to enhance presentation. Additionally, there is no annotation for the asterisks used to denote significance; it is assumed that everyone understands their meaning.

A number of the packages used in the manuscript, such as limma, have been published, but these have not been cited in the paper.

Experimental design

The research question is well-defined, relevant, and meaningful. However, the methods require more detail to ensure replicability.
- The manuscript states that five GEO datasets were combined into two (a test and a validation set) using random numbers. This is insufficiently detailed. The authors should describe the exact process. How was the data normalised across different datasets (GSE43458, GSE50081, etc.) before being combined? The use of SVA and limma is mentioned for batch calibration and normalisation, respectively, but the workflow should be more explicit.
The Western Blot protocol lacks some key details. The amount of total protein loaded per lane should be stated. The method of protein quantification (BCA kit) is mentioned, but not the final concentration or volume loaded.

Validity of the findings

The underlying data appears to be robust, sourced from public repositories. The statistical analyses employed (WGCNA, Cox regression, LASSO) are appropriate for the study's objectives. The division into a test set and a validation set is a major strength, adding credibility to the model's performance.

The conclusions are generally linked to the results, but some are overstated. The primary conclusion that the 6-gene signature has prognostic value for overall survival in LUAD is well-supported by the data in both the test and validation cohorts. However, the conclusion that "the model can be used to predict immunotherapy efficacy" is a significant overstatement. The authors show a correlation between the risk score and the immune microenvironment (e.g., T-cell infiltration scores, TME signals) and chemotherapy sensitivity, but they do not present any data on patient response to immunotherapy (e.g., checkpoint inhibitors). This claim must be toned down to suggest the model may have potential for predicting immunotherapy response, which requires further investigation. The current wording is not supported by the presented findings.

Additional comments

While the discussion covers the main findings, it could be enhanced by acknowledging the study's limitations more explicitly. For example, the model was built and validated on retrospective transcriptomic data and validated in vitro. Its clinical utility requires validation in a prospective patient cohort.

---

## Round 0.2 · accepted · Accept

· Academic Editor

Accept

Both the original reviewers admit that the revision has been done appropriately and now recommend its acceptance. Although Reviewer 2 still recommends clearly acknowledging the need for future prospective studies to validate the model's clinical utility, I believe that the manuscript can be accepted as is. If you would like to revise the manuscript again on this point, please let me know ASAP.

Reviewer 2 ·

Basic reporting

-

Experimental design

-

Validity of the findings

-

Additional comments

I have reviewed this revised manuscript and believe that the authors have satisfactorily addressed the concerns raised in the previous submission.

Reviewer 3 ·

Basic reporting

The manuscript has been rewritten in professional and clear English and observes the standard scientific article structure. The introduction provides sufficient background on lung adenocarcinoma (LUAD) and the tumour microenvironment, explaining why targeting regulatory T cells (Tregs) is a relevant research question. The authors appropriately cite prior literature to establish the context of their work. The figures and tables are relevant and appear to be labelled correctly. The authors state that all appropriate raw data have been made available, which aligns with the journal’s policy.

Experimental design

The research question is well-defined and meaningful. The study aims to fill a knowledge gap regarding the specific role of Tregs in LUAD and explore potential therapeutic targets. The methodology, which combines bioinformatics screening with experimental validation, is described in sufficient detail to be reproducible and corrects some of the shortcomings raised in my previous review.

Validity of the findings

The conclusions drawn are supported by the results presented. The study demonstrates that higher Treg expression is associated with a worse prognosis in LUAD patients, and the constructed prognostic model has predictive capabilities. The nomogram, which incorporates clinical factors and the risk score, further supports the model's clinical utility. The findings from the Western blot analysis, which showed increased expression of Treg markers and decreased expression of SCN9A in LUAD cells, are consistent with the bioinformatics analysis, strengthening the credibility of the results. However, the manuscript's conclusion notes a significant limitation: the study only shows a statistical correlation between gene expression and clinical outcomes, and the specific biological mechanisms require further experimental verification. This is a crucial point that should be more prominently acknowledged.

Additional comments

The manuscript provides a comprehensive analysis of a Treg-related prognostic model for LUAD. The structure is logical, from data acquisition and model construction to validation and biological pathway analysis.

The discussion section effectively points out the study’s strengths, such as integrating multiple datasets and validating the model, while also acknowledging the limitation of needing further mechanistic experiments. It's a good practice to explicitly state these points early in the discussion to provide a balanced view.

The study has potential, but should address the need for a prospective validation cohort to fully confirm the model's clinical efficacy, a point that is raised in the conclusion. This is a major limitation that impacts the direct clinical applicability of the findings. The current data, while promising, is based on a retrospective analysis.